# Involvement of Epithelial-Mesenchymal Transition (EMT) in Autoimmune Diseases

**DOI:** 10.3390/ijms241914481

**Published:** 2023-09-23

**Authors:** Julie Sarrand, Muhammad S. Soyfoo

**Affiliations:** Department of Rheumatology, Hôpital Erasme, Université Libre de Bruxelles, 1070 Brussels, Belgium

**Keywords:** epithelial to mesenchymal transition, fibrosis, chronic inflammation, innate immunity, adaptive immunity, rheumatic diseases, autoimmune disease, Sjögren’s disease, rheumatoid arthritis, systemic sclerosis, systemic lupus erythematosus, TGF-β, myofibroblast

## Abstract

Epithelial-mesenchymal transition (EMT) is a complex reversible biological process characterized by the loss of epithelial features and the acquisition of mesenchymal features. EMT was initially described in developmental processes and was further associated with pathological conditions including metastatic cascade arising in neoplastic progression and organ fibrosis. Fibrosis is delineated by an excessive number of myofibroblasts, resulting in exuberant production of extracellular matrix (ECM) proteins, thereby compromising organ function and ultimately leading to its failure. It is now well acknowledged that a significant number of myofibroblasts result from the conversion of epithelial cells via EMT. Over the past two decades, evidence has accrued linking fibrosis to many chronic autoimmune and inflammatory diseases, including systemic sclerosis (SSc), rheumatoid arthritis (RA), systemic lupus erythematosus (SLE), Sjögren’s syndrome (SS), and inflammatory bowel diseases (IBD). In addition, chronic inflammatory states observed in most autoimmune and inflammatory diseases can act as a potent trigger of EMT, leading to the development of a pathological fibrotic state. In the present review, we aim to describe the current state of knowledge regarding the contribution of EMT to the pathophysiological processes of various rheumatic conditions.

## 1. Introduction

Epithelial-mesenchymal transition (EMT) is a complex reversible biological process that takes place in epithelial cells and is characterized by the loss of epithelial and the acquisition of mesenchymal features [1,2]. Even though it was initially considered a binary process, a growing body of evidence indicates that EMT occurs in a stepwise manner with distinct states expressing different proportions of epithelial and mesenchymal markers. These intermediate states are defined as hybrid or partial EMT as they display attributes of both epithelial and mesenchymal cells [3,4,5]. In addition, the transitioning cells along the EMT spectrum keep the ability to revert at any moment to a more epithelial state by undergoing the reverse process known as mesenchymal–epithelial transition (MET) [6,7]. A broad variety of signals from the extracellular environment are able to trigger EMT in epithelial cells, causing dramatic alterations in their transcriptional and epigenetic landscapes, further eliciting the repression of epithelial junctions and the reorganization of the cytoskeleton [1,8,9]. As a result, cells lose their polarity, detach themselves from the basal membrane and adopt a spindle-shaped morphology similar to fibroblasts, thus increasing their motility and enabling the apparition of invasive properties [9,10,11].

EMT prevails in both physiological and pathological settings. It was initially described in the early 1980s by embryologists as a fine-tuned process that governs embryonic development [12]. It is now well established that EMT plays a pivotal role in several developmental processes including gastrulation, neural crest development, somite dissociation, and palate and lip fusion [13,14,15,16]. In adults, EMT can also be transiently activated as a physiological response to injury. It has been demonstrated that epithelial cells adjacent to the wound undertake EMT, allowing them to move while maintaining loose contact. Once the cells have arrived at the wounded site, epithelial barrier integrity is restored as the cells revert back to an epithelial phenotype via MET [17]. Unfortunately, EMT can also be erroneously triggered by more dramatic pathological conditions. Both organ fibrosis and neoplastic progression were found to be driven by EMT [1,10,18]. Over the past two decades, evidence has piled up to link EMT to various tumor functions, including tumor stemness, tumor cell migration, intravasation to the blood and lymphatic vessels, metastasis, and resistance to therapy [19,20,21,22]. Moreover, a great deal of attention has been paid to the contribution of EMT to chronic fibrotic diseases such as end-stage liver disease, chronic kidney disease, idiopathic pulmonary fibrosis (IPF), or heart failure [23,24]. Fibrosis is delineated by an excessive number of myofibroblasts, resulting in exuberant production of extracellular matrix (ECM) proteins (i.e., collagen and fibronectin), thereby compromising organ function and ultimately leading to organ failure [25,26,27]. It is now well acknowledged that a significant percentage of myofibroblasts result from the conversion of epithelial cells via EMT [28,29]. As fibrosis has been linked to many rheumatic diseases, including systemic sclerosis (SSc) [30], rheumatoid arthritis (RA) [31], systemic lupus erythematosus (SLE) [32], and Sjögren’s syndrome (SS) [33], it is therefore suspected that chronic inflammatory states could act as a potent trigger of EMT, leading to the development of a pathological fibrotic state in these rheumatic diseases [1,4,10,34,35]. 

Even though several lines of evidence seems to confirm the occurrence of EMT in rheumatic diseases [36,37], its precise contribution to their pathophysiology remains unclear. In the present review, we therefore aim to describe the current state of knowledge regarding the contribution of EMT to the pathophysiological processes of various rheumatic autoimmune diseases. 

## 2. Biological Mechanisms Underlying EMT

EMT is a tremendously complex biological process. Despite the quantum leaps made to unearth the different regulatory mechanisms underlying EMT, the exact regulation of this process is not yet fully understood. A broad range of signals stemming from the stromal microenvironment bind to their cognate receptors on epithelial cells and are able to trigger several distinct intracellular signaling pathways which can display cross-talk to some extent. These intracellular signaling events ultimately converge on activating EMT via the expression of core EMT-transcription factors (TFs) in various combination that lead to the repression of epithelial genes and the activation of mesenchymal ones, accounting for the loss of epithelial and the gain of mesenchymal traits. In addition, a process similar to EMT has been observed and described in cell types other than epithelial cells, suggesting that the frontiers of EMT might be looser than previously thought and that EMT might not be restricted only to epithelia cells [38]. This EMT-like process has been widely described in non-epithelial cancers such as melanoma, sarcoma, and leukemia [39,40,41] and in non-cancerous conditions where it aroused in non-epithelial cells such as endothelium [42], mesothelium, [43] or fibroblast-like synoviocytes (FLSs) [44]. Several authors have proposed other terminologies to avoid confusion with EMT arising in epithelial cells and called these processes either endothelial–mesenchymal transition (endoMT) [42] for endothelial cells, mesothelial–mesenchymal transition (MMT) [43] for mesothelial cells, or EMT-like for other cell types than epithelial cells [45].

### 2.1. Morphological Modification during EMT

Epithelial cells are closely held together by different types of lateral cell-cell junctions (i.e., adherens junctions, desmosomes, gap junctions and tight junctions). Epithelial cells form polarized sheets and harbor an apico-basal polarity by anchoring to the basement membrane via hemidesmosomes, which contain α6β4 integrin, and are linked to cytokeratin intermediate filaments, a compound of the cytoskeleton enabling epithelial cells to resist numerous physical stresses, inside the cell [11,19]. The loss of E-cadherin (a key component of adherens junctions) is often considered a prototypical event of EMT and is rapidly followed by the repression of other cell junction proteins such as zona occludens 1 (ZO-1), occludin, and claudin [46,47,48]. Consequently, the dissolution of epithelial cell junctions elicits a loss of apico-basal polarity and triggers cytoskeletal modifications with a switch from cytokeratin-rich to vimentin-rich networks, leading to the adoption of a spindle-shaped morphology [48]. These events permit cells to gain mobility and to migrate into surrounding tissues along a secreted matrix of fibronectin [10]. As mentioned before, the cells are pushed towards the EMT spectrum in a progressive and reversible manner, from a complete epithelial to complete mesenchymal state ultimately characterized by mesenchymal markers, such as vimentin, fibronectin, α-smooth muscle actin (α-SMA), N-cadherin, fibroblast specific protein 1 (FSP-1), and various matrix metalloproteases (MMPs) and the loss of epithelial markers such as E-cadherin, ZO-1, occludin, and several cytokeratins (Figure 1) [10,11,19].

### 2.2. Transcriptional Regulation of EMT

EMT is an elaborate process that is tightly regulated at the transcriptional level. Over the years, serious efforts have been made to disentangle this complex system. It has become clear that a great deal of transcription factors (TFs) are involved in the gene regulatory networks controlling EMT. Some of these have identified as master regulators or core EMT-TFs, as they are conserved across various biological conditions and across species, whereas other TFs may have more restricted functions depending on the tissue or the biological background [49,50], suggesting that the transcriptional regulation of EMT involves multiple layers of control [20,50]. Consequently, EMT core-TFs tend to control one another’s expression along with additional TFs that further define and drive EMT progression. Altogether, EMT-TFs act synergistically with one another, using some common pathways to coordinate the repression of epithelial and activation of mesenchymal phenotypes [51]. Core EMT-TFs include the zinc-finger E-box-binding homeobox factors zinc finger E-box binding homeobox 1 (ZEB1), ZEB2, snail family transcriptional repressor 1 SNAI1 (also known as SNAIL), SNAI2 (also known as SLUG), and the basic helix–loop–helix factors (bHLH) twist family BHLH transcription factor 1 (TWIST1) and TWIST2 [52,53].

#### 2.2.1. SNAI Family 

SNAI1 and SNAI2 belong to the SNAIL family of zinc-finger TFs. SNAI1 and SNAI2 are potent repressors of E-cadherin (CDH1) gene transcription as they bind to its promoter, where they recruit several corepressor complexes [54,55,56]. SNAI1 is also involved in the repression of multiple epithelial genes in charge of the maintenance of epithelial phenotype, including genes encoding claudin and occludin, major components of tight junctions [57]. In addition, SNAI1 cooperates with TWIST1 in the induction of ZEB1 expression [58] and with ETS1, itself ultimately activated by mitogen-activated protein kinase (MAPK) signaling, to induce MMP expression, a large family of endopeptidase involved in ECM remodeling [59]. SNAI1 can also interact with the small mothers against the decapentaplegic homolog (SMAD)3–SMAD4 complex, further entailing the transformation of growth factor (TGF)-β-mediated repression of E-cadherin and occludin [60]. 

#### 2.2.2. ZEB Family

ZEB1 and ZEB2 are the two members of the ZEB family of zinc finger TFs. Similar to SNAI1, ZEB1 binds to CDH1 promoter and represses transcription via the recruitment of corepressor complexes [61,62]. ZEB1 mediate the transcriptional activation of the genes encoding vimentin and N-cadherin via the recruitment of coactivator complexes to their respective promoters [49,63]. Additionally, ZEB2 together with SNAI1 increase the expression of MMPs [64,65]. 

#### 2.2.3. BHLH Family 

TWIST1 and TWIST2 both belong to the bHLH class of transcriptional regulators. As with SNAI1 or ZEB1, TWIST1 suppresses the expression of the epithelial gene CDH1 and activates the expression of N-cadherin, a mesenchymal gene [66] due to the recruitment of co-repressor complexes at promoter sites [67]. Additionally, TWIST1 expression can be induced under hypoxic conditions by hypoxia-inducible factor 1α (HIF1α), further promoting EMT [68].

#### 2.2.4. Epigenetic Regulation

Epigenetic modifications have been newly discovered to play a crucial role in EMT related to neoplastic and fibrotic diseases, adding even more complexity to this fine-tuned process. Epigenetic modifications encompass multiples processes including histone deacetylation, lysine methylation, long noncoding ribonucleic acids (lncRNAs) and micro-RNA (miRNAs). The miR-200 family is among the most studied miRNAs in cancer and IPF [69,70]. miRNAs are a class of small (17–22 nucleotides) non-coding RNA molecules that negatively regulate gene expression by targeting messenger RNA (mRNA) resulting from gene transcription [71]. The miR-200 family is among the most studied miRNAs in cancer and IPF and inhibits the expression of ZEB factors [72,73]. In turn, ZEB1 and ZEB2 bind to the promoters of miR-200, thereby allowing for a reciprocal feedback loop controlling EMT [72,74]. Evidence showed a dramatic reduction of miR-200 in cells undergoing EMT [69,70]. 

### 2.3. Signaling Pathways Regulating EMT 

TGF-β signaling is considered as the prototypical pathway involved in EMT and has been extensively studied. However, EMT can also be induced via several other intracellular signaling pathways resulting from the encounter between epithelial cells and specific ligands. These pathways are not well compartmentalized, and some overlap exists to a certain extent between them (Figure 2).

#### 2.3.1. TGF-β: SMAD-Dependent and SMAD-Independent Signaling Pathways

TGF-β1 is a member of the TGF-β superfamily of cytokines and signals via a tetrameric complex consisting of two TGFβ receptor type I (TβRI) and two TGFβ receptor type II (TβRII), a transmembrane serine-threonine kinase receptor. The TGF-β1 intracellular signaling pathway is subdivided into a canonical SMAD-dependent and a non-canonical SMAD-independent pathway [75]. 

The binding of TGF-β1 to its receptor leads to the phosphorylation and activation of cytoplasmic SMAD2 and SMAD3, which further assemble with SMAD4 to migrate to the nucleus, where they activate the transcription of several EMT-TFs such as SNAI1, SNAI2, TWIST1, and ZEB1 [66,76]. SMAD6 and SMAD7 are negative regulators of TGF-β1 signaling and prevent SMAD2 and SMAD3 from binding to TβRI [77,78]. Additionally, it has been demonstrated that SMAD proteins interact with other signaling pathways such as WNT and NOTCH pathways via their interaction with β-catenin and NOTCH receptor intracellular domain (NOTCH-ICD) [79].

TGF-β1 can also activate non-SMAD signaling pathways. It activates several signaling pathways such as phosphoinositide 3-kinase (PI3K)–protein kinase B (AKT), extracellular signal-regulated kinase (ERK) MAPK, p38 MAPK, and c-Jun N-terminal kinase (JNK) pathways [11,80,81]. Several studies have established that TGFβ1 enables the activation of p38 MAPK and JNK pathways following the association of TNF receptor-associated factor 6 (TRAF6) with the TGF-β receptor complex [82,83]. In addition, TGF-β is also competent to activate the mammalian target of rapamycin (mTOR) complex [81] and nuclear factor-κB (NF-κB) via its collaboration with the PI3K–AKT pathway [84]. All those pathways ultimately converge in the activation of core EMT-TFs, further entailing the activation of mesenchymal genes and the repression of epithelial ones. 

#### 2.3.2. WNT Signaling Pathway

The WNT/glycogen synthase kinase-3β (GSK-3β)/β-catenin axis is an evolutionarily conserved signaling pathway that plays a pivotal role in cellular homeostasis and regulates miscellaneous processes including embryological development, cell proliferation, differentiation, apoptosis, and EMT [85]. WNT ligands bind to and activate Frizzled receptors and low-density lipoprotein receptor-related protein (LRP5/6), which promote the inhibition of GSK-3β via the cytosolic proteins disheveled (DVL) [86]. This enables the release of β-catenin from the GSK3β-axis inhibition protein (AXIN)–adenomatous polyposis coli protein (APC) complex and its translocation to the nucleus, thereby activating T cell factor (TCF) and lymphoid enhancer binding factor 1 (LEF-1) TFs, which promotes the expression of various EMT-associated genes such as SNAI1 [87,88].

#### 2.3.3. NOTCH Signaling Pathway

NOTCH signaling is a highly conserved intercellular communication pathway involved in developmental processes, cancer, and fibrosis [89,90,91]. Delta-like (DLL)1, DLL3, DLL4, Jagged (JAG)1, and JAG2 are the cognate ligands of the NOTCH receptors (NOTCH 1–4). Upon activation of the NOTCH receptor, γ-secretase mediates the cleavage of the active NICD, ensuing its migration to the nucleus to function as a transcriptional co-activator that complex with CSL (CBF-1/Suppression of hairless/Lag1), a DNA-binding protein, to stimulate the expression of EMT TFs [92,93]. In addition, NOTCH signaling can regulate expression of *SNAI2* both directly [94,95] and indirectly via the induction of HIF-1α [96]. NOTCH-ICD can also stabilize cytoplasmic β-catenin and activate other pathways, such as ERK and NF-κB, that induce the SNAI1/2 and LEF-1 TFs [97,98,99].

#### 2.3.4. Hedgehog Signaling Pathway

Hedgehog signaling is conserved across species and plays a critical role in embryonic development [100,101]. In adults, it was linked to EMT in various cancers and fibrotic diseases [102,103]. Members of the Hedgehog (HH) ligand family include Sonic (SHH), Desert Hedgehog (DHH) and Indian Hedgehog (IHH) [104], which bind to their transmembrane protein receptors, patched (PTCH)1 and PTCH2 [105]. The activation of PTCH1/2 releases Smoothened (SMO), which ultimately leads to the activation of GLI TFs, fostering the transcription of target genes such as PTCH, WNT, and SNAI1 [106,107,108]. 

#### 2.3.5. Receptor Tyrosine Kinases (RTKs) Signaling Pathway

Receptor tyrosine kinases (RTKs) belong to a large family of cell surface receptors that play a significant role in cell–cell communication, controlling complex biological functions, including cell growth, motility, differentiation, and metabolism [109,110]. Since the discovery of the first RTK in the early 1990′s, a wide variety of growth factors, cytokines, and hormones have been identified as ligands of their cognate RTKs, triggering receptor dimerization followed by receptor activation via its auto-phosphorylation. This in turn enables the activation of the PI3K–AKT, ERK–MAPK, p38 MAPK, and JNK pathways [111]. In particular, growth factors such as epidermal growth factor (EGF), fibroblast growth factor (FGF), hepatocyte growth factor (HGF), and vascular endothelial growth factor (VEGF) have been identified as potent inducers of EMT [112,113,114]. These signaling cascades activate TFs that bind to the promoters of core EMT-TFs, such as SNAI1/2, ZEB1/2, and TWIST1 [115,116]. Additionally, RTK signaling displays some cooperation with other distinct pathways. For example, AKT activation can either induce SNAI1 expression via NF-κB activation [117] or β-catenin activation via inhibition of GSK3β of WNT signaling [118]. TGF-β signaling can also enhance EMT responses initiated by growth factors such as FGF or EGF via the SMAD-independent pathway [119]. 

#### 2.3.6. JAK/STAT Signaling Pathway

Janus kinase (JAK)/signal transducers and activators of transcription (STAT) signaling is critical in regulating embryonic development and inflammatory responses and has been linked to malignancies and autoimmune diseases [120,121,122]. The JAK family consists of intracellular non-receptor tyrosine kinase proteins that are activated by several cytokines or growth factors, ensuing the dimerization and activation of the transcription protein STATs. Several lines of evidence have shown that STAT dimers activate the transcription of genes encoding EMT-TFs (i.e., SNAI1, ZEB1, JUNB, and TWIST1) [123,124,125].

### 2.4. Extracellular Elements Regulating EMT

#### 2.4.1. Inflammation

It is now well recognized that inflammation contributes to EMT initiation. Proinflammatory cytokines such as tumor necrosis factor (TNF)-α, and interleukin (IL)-1 are potent inducers of the NF-κB pathway entailing the secretion of various cytokines, chemokines, and adhesion molecules, all of which contribute to the recruitment of immune cells, further perpetuating a pro-inflammatory environment [126]. Additionally, several studies have linked the activation of the NF-κB pathway to EMT via its direct activation of core EMT-TFs including SNAI1, ZEB1, and ZEB2 [117,127,128,129]. SNAI1 can in return activate the transcription of pro-inflammatory cytokines genes as a positive feedback loop response [3,130,131]. In cancer and fibrotic diseases, IL-6 and IL-17 were able to promote EMT via the JAK-STAT signaling pathway and induce SNAI1 expression [132,133,134]. In addition, IL-17 has been shown to promote EMT in various conditions such as lung cancer, IBD, and IPF [135,136,137]. In particular, evidence suggests that IL-17 induces EMT in A549 alveolar epithelial cells via the TGF-β mediated SMAD signaling pathway [137].

#### 2.4.2. Hypoxia-Induced EMT

Hypoxia is considered an EMT permissive state and has been linked to numerous conditions such as cancer, fibrosis, and autoimmune diseases [138,139,140]. Under hypoxic conditions, prolyl hydroxylase (PHD) inactivation lead to HIF-1α accumulation, further driving expression of core EMT-TFs such as SNAI1, TWIST1, ZEB1, and ZEB2 [68,141,142,143]. Furthermore, evidence has shown that hypoxia is a potent inducer of the TGF-β SMAD-dependent and PI3K/AKT pathways via HIF-1α accumulation, causing the apparition of EMT in renal fibrosis and IPF, respectively [144,145]. It was also linked in several studies with the TGF-β SMAD-dependent [146], NF-κB [147], and NOTCH signaling pathways [96].

#### 2.4.3. Extracellular Matrix Induced EMT

ECM itself plays an important part in EMT. EMT generates multiple modifications to the composition and the mechanical properties of ECM, and in return, ECM influences EMT in a feedforward manner. These interactions have mostly been studied in many areas of cancer research field, as ECM remodeling and increased matrix stiffness observed during EMT has been linked to tumor invasion and metastasis [148]. Type I collagen enables EMT via the binding and activation of receptors such as discoidin domain receptors (DDR)1, DDR2 and integrins alpha 2 beta 1. Those receptors activate NF-κB that ultimately promote expression of SNAI1/2 and LEF1 [149,150,151]. In addition, MMPs, a large family of endoproteases that promote the degeneration of various components of the ECM, have been linked to EMT through their capacity of remodeling of ECM further facilitating migration and invasion of cells. Conversely, the remodeling of ECM by MMPs can also modulate EMT by activating several signaling pathways from the ECM and modulating the bioavailability of several ligands [152]. More specifically, MMP-3 has been shown to directly induce EMT via the activation of Rac1 GTPase–reactive oxygen species (ROS) signaling, which promotes expression of SNAI1 and further supports EMT progression [153]. 

## 3. EMT in Various Rheumatic Diseases

### 3.1. Rheumatoid Arthritis

RA is a chronic systemic inflammatory autoimmune disease that primarily affects joint inflammation, pain, and stiffness and can, if left untreated, lead to irreversible joint destruction and functional disability [154]. The pathophysiology of RA arises from complex interactions with multiple genetic, environmental, and immunological factors, further ensuing a breakdown of immune tolerance and the development of chronic synovial inflammation [155]. Despite the progress made in the understanding of the disease, the exact mechanisms underlying RA physiopathology still remain elusive to this day. Current treatment strategies targeting inflammation involve traditional disease-modifying anti-rheumatic drugs (DMARS) as well as biologic agents that target pro-inflammatory cytokines, including TNF-α or IL-6, B cells, or the activation of T cells. Despite this multiplicity of drugs, several observational studies still reported 16 to 21% of patients with refractory disease, suggesting the need for treatments targeting pathways other than inflammation [156].

One of the most preeminent features of RA is the striking morphological and functional transformations occurring in the synovium, including synovial lining hyperplasia and inflammatory cell infiltration. These modifications can ultimately lead to the formation of an invasive tumor-like tissue called pannus that enables invasion of the periarticular bone at the cartilage–bone junction, further hastening the development of bone erosion and cartilage destruction [157,158]. The pannus is constituted of synovial tissue hyperplasia when the lining size increases from a physiological one to three cells layers up to 10–15 cell layers [159] in which neovascularization processes [160] are observed together with the accumulation of inflammatory infiltrates including T cells, macrophages, B cells, plasma cells, dendritic cells, and mast cells [161,162]. RA fibroblast-like synoviocytes (RA-FLSs) mainly account for this significant tissue hyperplasia [163,164] and share many biological properties with tumor cells, including tumor-like proliferation, migration, and invasion as well as an increased resistance to apoptosis, even in the absence of inflammatory stimulus [165,166,167]. In addition, aberrantly activated RA-FLSs exhibit promigratory adhesion molecules, upregulated expression of proto-oncogenes and increased production of pro-inflammatory cytokines and MMPs (Figure 3) [168,169].

Over the past 15 years, several lines of evidence seem to indicate that RA-FLSs may, at least in part, undergo EMT-like process (Table 1) [31,170,171]. This conclusion was indeed supported by immunohistological analysis of both healthy and RA synovium that showed the presence of α-SMA, a myofibroblast marker responsible for collagen accumulation in fibrosis exclusively in the synovial lining layer of RA patients [31,171,172]. Furthermore, exposure to synovial fluid from RA patients induced the expression of α-SMA in healthy FLSs, suggesting the presence of an EMT-like process. Additionally, concomitant upregulation of α-SMA and podoplanin, a glycoprotein physiologically expressed in the lymphatic endothelium and associated with EMT in invasive cancers [172,173], were found in RA-FLS, whereas both markers were absent in osteoarthritis (OA) FLS (Figure 3). In vitro treatment with pro-inflammatory cytokines known to promote EMT-like behavior (i.e., IL-1β, TNF-α, and TGF-β1) induced expression of both α-SMA and podoplanin in OA-FLS, thus evoking the role of podoplanin in the activation of RA-FLS leading to their increased migratory potential [172]. Mechanistic in vitro studies showed that TGF-β1 enhanced migration and invasion by inducing EMT-like processes via the activation of SMAD2/3 in RA-FLSs, which led to the upregulation of core EMT-TF SNAI1. The inhibition of SMAD2/3 blocked EMT and limited the migration and invasion induced by TGF-β1, therefore suggesting that the TGF-β1 SMAD-dependent signaling pathway is involved in EMT-like process and contributes to migration and invasion of RA-FLSs [31]. Moreover, several studies evaluated the effect of hypoxia on RA-FLSs, as hypoxia is a well-defined inducer of EMT and is commonly found in synovial tissues of RA patients [174]. These studies suggested that the activation of the PI3K/AKT/HIF-1α and the NF-κB/HIF-1α pathways plays a significant role in hypoxia-induced EMT-like process occurring in RA-FLSs [44,175]. HIF-1α was also shown to directly regulate the expression of NOTCH-1 and NOTCH-3 under hypoxic conditions, further mediating the activation of RA-FLSs [176]. In addition, a synergetic effect of IL-17A and hypoxia was demonstrated via upregulation of the expression of MMP2 and MMP9 via the activation of the NF-κB/HIF-1α pathway [175]. The use of Baicalein, an anticancer drug inducing cancer cell apoptosis, successfully inhibited FLS proliferation and migration via inhibition of the PI3K/Akt/mTOR pathway in a human synovial sarcoma cell line (SW982), a model of in vitro RA-FLSs. The upregulation of E-cadherin together with the downregulation of vimentin, MMP2, MMP9, and SNAI1 suggests that Baicalein promotes MET, the reverse process of EMT [177]. A pre-clinical study using collagen-induced arthritis (CIA) in rats also identified Snail as a potent inhibitor of Phosphatase and TENsin homolog (PTEN) and activator of Pi3k-Akt pathways and was linked to the development of the invasive phenotype of synovial cells [178]. CircCDKN2B−AS_006, a circular RNA, was found to be upregulated in the synovial biopsies from RA patients. A pre-clinical study on CIA rats confirmed the role of circCDKN2B−AS_006 in the proliferative and invasive phenotype of RA-FLSs, whereas its knockdown alleviated the severity of arthritis. Mechanistic analysis concluded that circCDKN2B−AS_006 promoted EMT-like processes in RA-FLSs by inducing the expression of runt-related transcription factor 1 (Runx1), ultimately activating the Wnt/β−catenin signaling pathway (Table 1) [170].

### 3.2. Sjögren’s Syndrome

SS is a chronic systemic autoimmune disease defined by lymphoplasmacytic infiltration of the exocrine glands subsequently leading to xerostomia and keratoconjunctivitis sicca, also referred as sicca syndrome. The disease is classified either as primary SS (pSS) when it occurs in the absence of other autoimmune diseases or secondary SS (sSS) when associated with other underlying autoimmune disorders such as SLE, RA and SSc [179]. Beyond the typical sicca syndrome, systemic manifestations resulting from the lymphocytic infiltration of organs can be present in up to 40% of cases including ILD, neurological manifestations and even an increased risk of lymphoma [179,180]. Over the past decades, several lines of evidence have pointed toward the pivotal role of salivary gland epithelial cells (SGECs) in the pathogenesis of pSS, especially in the initiation and progression towards autoimmunity, justifying the increasingly widespread use of the term “autoimmune epithelitis” [181]. There are consistent data indicating that the conjuncture of genetic predisposal, environmental insults, and hormonal disequilibrium may lead to the activation of the resting epithelium, the upregulation of toll-like receptors (TLRs) such as TLR-2, 3, 4, 7, 8, and 9 [182,183,184,185], leading to the release of alarmins and pro-inflammatory cytokines such as interferon (IFN), TNF-α, IL-6 and IL-17 further promoting downstream autoimmune inflammation [184,186], ultimately resulting in the atrophy and fibrosis of the salivary glands (SG) (Figure 4) [187,188].

It was demonstrated that SG fibrosis is positively correlated with biopsy focus score and ocular surface damage [189], which led to the prevailing hypothesis suggesting that chronic exposure to pro-inflammatory mediators may drive SG fibrosis via an EMT-dependent fibrotic process (Table 2) [190,191]. Indeed, compared to SG samples from healthy subjects, immunohistochemical staining analysis of SG from SS patients consistently showed the presence of SGECs with a myofibroblast-like phenotype characterized by a spindle-shaped morphology; a strong positivity for mesenchymal markers such as vimentin, collagen type I, SNAI1, and phosphorylated SMAD2/3 and SMAD4; and a decreased expression of the epithelial marker E-cadherin [33,192]. More interestingly, a significant increase in podoplanin was also found in immunostaining of SGs from pSS patients and was associated with higher numbers of lymphatic capillaries, as well as with the presence of myoepithelial cells neighboring the ductal epithelium [193]. In addition, upon exogenous TGF-β1 treatment, SGECs from healthy donors cultured in vitro were able to recapitulate the abnormalities observed in pSS SG samples, including increased gene expression of mesenchymal markers such as phosphorylated SMAD2/3, SMAD4, SNAI1, vimentin, and collagen type I and the reduction of E-cadherin expression. In contrast, the use of the specific TβR1 inhibitor SB-431542 in healthy SGECs treated with TGF-β1 showed a significant reduction in vimentin and collagen type I expression and E-cadherin levels similar to healthy SGECs [33]. Other studies highlighted the contribution of other cytokines such as IL-6, IL-17, and IL-22 to the EMT-dependent fibrotic process taking place in SGs of SS patients. Treatments of primary cultures of SGECs obtained from healthy donors with IL-6, IL-17, and IL-22 enabled all prototypical changes related to EMT, such as the apparition of a spindle-shaped morphology, together with the upregulation of vimentin and collagen type I and downregulated E-cadherin [191,192,194]. A mechanistic study confirmed the role of IL-17 in the activation of the canonical TGF-β1/SMAD2/3 and the noncanonical TGF-β/ERK1/2 pathways in human SGECs derived from healthy subjects. Additionally, co-treatment with IL-17 and specific TβR1 inhibitor SB431542, or ERK1/2 inhibitor U0126, abrogated the corresponding morphological changes and EMT markers expression in healthy SGECs [194]. More recently, ETS1 and MMP9 were found to be overexpressed in SGECs from patients with pSS. Further functional analysis demonstrated that ETS1 directly induced expression of MMP9 by exercising its TF activity and affected the expression of EMT markers in immortalized SGECs (iSGECs) from pSS [195]. In addition, the presence of EMT in SGEC from pSS patients was further strengthened by RNA sequencing data showing an over-representation of a pathway involving MMPs (collagen type III Alpha 1 chain (COL3A1), COL1A2 and A Disintegrin And Metalloproteinase with Thrombospondin Motifs (ADAMTS)) and core-EMT TF such as ZEB2 (Table 2) [196].

### 3.3. Systemic Lupus Erythematosus 

SLE is a chronic autoimmune connective tissue disease that affects mainly young woman of childbearing age [197]. It is defined by a chronic and excessive immune response against self-antigens resulting in the production of autoantibodies and immune complexes (IC), further causing inflammation and tissue damage in multiple organs such as the joints, skin, kidneys, lungs, and brain [198]. Lupus nephritis (LN) is a common, yet devastating complication which affects 20–60% of SLE patients over time and is still associated with significant morbidity and mortality [199]. Even though LN is usually characterized by glomerulonephritis, it is frequently associated with tubular, interstitial, and capillary lesions [200] with reported incidence of tubulointerstitial fibrosis as high as 50% [201], which led to the suspicion of the presence of an EMT-dependent fibrosis process in tubular epithelial cells derived from human renal biopsies collected from LN [202]. Interestingly, two preclinical studies investigated the origin of interstitial α-SMA fibroblasts by using kidney fibrosis mouse model and concluded that EMT programming contributed to the generation of fibroblasts in 5% and 36% of patients, respectively [29,203]. 

It is now well established that TGF-β1 is a potent inducer of pathological EMT-dependent fibrosis occurring during chronic kidney disease [202,204] via the downregulation of pro-epithelial markers, including E-cadherin and ZO-1; the production of MMP2 and MMP9, which alter the tubular basement membranes’ integrity and induce marked morphological changes accompanied by the de novo synthesis of the mesenchymal marker α-SMA; and the acquisition of migratory capacities facilitating the invasion of the interstitium (Table 3) [205,206]. Studies have demonstrated a significant increase in the immune expression of TGF-β1 and TGF-β3 in renal biopsies of patients with LN [207,208]. It is suspected that anti-double-stranded DNA antibodies bind to mesangial cells, leading to the secretion of pro-inflammatory cytokines and to an increase in the synthesis of profibrotic TGF-β1 and fibronectin [208,209]. In addition, limited studies have demonstrated the potential involvement of the Th17/IL-17 axis in EMT arising in tubular epithelial cells [210,211]. In cultured proximal tubular epithelial cells (HK-2 cell line), IL-17A treatment elicited the expression of collagen I and III and α-SMA and the loss of E-cadherin via a TGF-β1-dependent pathway [210]. miR-130b-3p, a member of miRNA family, was significantly upregulated in serum of patients with early stage LN compared to healthy controls and positively correlated with 24-h proteinuria and renal chronicity index. Upon stimulation of TGF-β1, miR-130b-3p transfected renal tubular cell line (HK-2) enable the decrease of E-cadherin expression and increase of α-SMA, whereas the opposite effects were obtained with miR-130b-3p inhibitors therefore preventing TGF-β1 induced EMT [212]. Recently, neutrophil extracellular traps (NETs) were more abundantly found in lung biopsies of SLE patients compared to controls. RNA sequencing arising from these samples indicated that NETosis promotes EMT in lung epithelial cells from SLE patients. These data were further confirmed in vitro by demonstrating that stimulation with NETs significantly upregulated the expression of α-SMA, Twist1, and Snai1 while downregulating the expression of E-cadherin [213]. In addition, NET was also identified in renal biopsies from patients with active LN and correlated with the severity of proteinuria and with glomerular endoMT. These results were further investigated in cultured endothelial cells (ECs), where NET could induce endoMT through the degradation of vascular–endothelial (VE)–cadherin, leading to the activation of β-catenin signaling, which ultimately resulted in an increased expression of SNAI1, MMP9 and α-SMA. In MRL/lpr mice, NET also correlated with proteinuria and was associated with the glomerular upregulation of endoMT markers, including Snail1 and MMP9 (Table 3) [214].

Several pre-clinical studies were performed using the lupus-prone model of MRL/Lpr mice and underlined the role of several molecules in EMT-dependent fibrosis and their potential interest as new therapeutic targets. As an example, targeting oncostatin M (OSM), a member of the IL-6 cytokine family highly expressed in renal tissue of LN mice, improved tubulointerstitial fibrosis with increase expression of E-cadherin and decrease of mesenchymal markers such as α-SMA, fibronectin (FN) and Stat1/Stat3, indicating that OSM may activate EMT via the Jak-Stat signaling pathway [215]. Similarly, treatment targeting fractalkine (FKN), a chemokine (C-X3-C motif) ligand 1 known to be involved in the disease progression of LN, also improved renal function and alleviated tubule-interstitial fibrosis. Mesenchymal markers such as α-SMA and vimentin were reduced, together with members of the Wnt signaling pathways, including Wnt-4 and β-catenin. Those results were confirmed in human FKN-depleted HK-2 cell line. In addition, inhibition of the Wnt/β-catenin pathway by XAV939 alienated the effects of FKN overexpression [216]. In addition the expression of Stim1 in the kidney tissues of LN mice was significantly increased compared to control mice and was positively correlated with fibronectin, urine protein, and serum creatinine levels [217]. Yap1 expression was also significantly higher in the renal tissues of MRL/lpr mice compared to normal mice, and was positively correlated with FN and with the degree of renal function injury [218]. Recently, treatment with iguratimod, an anti-inflammatory small molecule drug approved in Japan for treatment of RA that lowers the production of various cytokines including TNF-α, IL-1β, IL-6, IL-8 and IL-17, successfully alleviated tubulo-interstitial lesions and fibrosis using MRL/lpr mice. In addition, in vitro co-treatment with TGF-β1 and iguratimod impeded morphological changes associated with fibrosis, downregulated E-cadherin and upregulated fibronectin [219]. Jieduquyuziyin, a traditional Chinese medicine used for LN, displayed therapeutic actions in MRL/lpr mice, where it improved renal function and alleviated renal fibrosis. Kidney analysis of Jieduquyuziyin-treated mice showed E-cadherin upregulation together with the downregulation of vimentin, α-SMA, TGF-β1 and phosphorylated Smad2/3. These data suggest that the therapeutic effect of Jieduquyuziyin could be related to the inhibition of EMT due to the inhibition of the TGF-β1/Smad2/3 signaling pathway (Table 3) [220].

### 3.4. Systemic Sclerosis

SSc is a rare and heterogeneous autoimmune connective tissue disease characterized by fibroproliferative vasculopathy, auto-immunity and aberrant fibrosis of the skin and internal organs, causing progressive impairment and failure of the affected organs. Even though the complexity of the disease translates into a wide spectrum of clinical manifestations, SSc is classically divided in two distinct subsets based on the pattern of skin involvement [221,222]. Limited cutaneous SSc (lcSSc) is usually characterized by limited skin fibrosis and vascular manifestations, whereas diffuse cutaneous SSc (dcSSc) is dominated by a rapidly progressing and widespread fibrosis of the skin and internal organs, often associated with a disastrous prognosis despite the considerable progress made in the management of patients and the understanding of the underlying pathophysiology [223]. The prevailing hypothesis suggests the presence of a complex interplay between vascular cells, immune cells and fibroblasts. In individuals with a permissive genetic background, triggering events such as viral infection or exposure to environmental toxins lead to endothelial cell damage and apoptosis, thus resulting in the release of pro-inflammatory cytokines, damage-associated molecular patterns (DAMPs) and ROS and the subsequent activation of the innate and adaptive immune system [224,225,226]. Activated immune cells lead to these release of a wide variety of pro-inflammatory and pro-fibrogenic cytokines such as IL-4, IL-6, IL-13, TNF-α and TGF-β together with the secretion of autoantibodies by activated B cells, entailing further tissue damage and the apparition of activated myofibroblasts [222,227]. These activated myofibroblasts, which are normally absent in healthy tissues, display a marked resistance to apoptosis, enhanced ECM synthesis, constitutive secretion of pro-fibrotic cytokines and are suspected to arise from a wide variety of cells ranging from resident fibroblasts to pericytes, adipocytes, endothelial and epithelial cells through processes of transdifferentiation including EMT and endoMT [228,229,230].

The presence of EMT in skin and lungs of SSc patients is supported by several lines of evidence (Table 4). Firstly, treatment with TGF-β and/or TNF-α, two well-known inducers of EMT which are overexpressed in SSc patients, were able to induce EMT in normal human epidermal keratinocytes (NHEK cells) and was characterized by morphological changes including apparition of spindle-shaped cells, downregulation of epithelial markers such as E-cadherin and ZO-1, and upregulation of mesenchymal markers such as vimentin, fibronectin, and MMPs. Treatment with SB431542, a SMAD inhibitor, significantly prevented EMT and could even reverse established EMT phenotype [231]. In addition, immunostainings performed on skin biopsies of patients with morphea, a form of localized scleroderma, suggested the involvement of EMT based on the upregulation of mesenchymal markers such as TGF-β1, α-SMA, fibronectin and SNAI1 and the downregulation of E-cadherin [232]. The expression of core EMT-TFs such as SNAI1 and TWIST1 was also reported in biopsies of patients with dcSSc [233]. Furthermore, skin biopsies from dcSSc patients showed increased expression of phosphorylated SMAD2/3 (active form) and SNAI1, suggesting that the TGF-β SMAD-dependent signaling pathway is involved in the fibrotic skin of dcSSc patients [230]. Similarly, human alveolar epithelial cells type II exposed to TGF-β also displayed the expression of EMT markers and acquired the capacity to produce collagen [234]. Several other EMT-related pathways, such as the WNT and NOTCH pathways have been shown to be involved in SSc [235,236,237]. Micro-RNA 138 (MiR-138), a well-established repressor of EMT in several types of cancers [238], was found to be significantly decreased in the sera of patients with dcSSc and lcSSc compared to healthy controls, further pointing to the involvement of EMT in SSc pathophysiology [239]. In addition, the presence of an EMT-like process arising in endothelial cells, which is also referred to as endoMT, has also consistently been described in skin tissues taken from SSc patients [240,241,242,243]. Immunohistochemical analysis of skin biopsies from SSc showed significant upregulation of α-SMA, HIF-1α and VEGF-α compared with healthy controls. These observations were further reproduced by culturing human endothelial cells under hypoxic conditions [240]. Lymphatic endothelial cells were also linked with endoMT during histological analyses of skin sections from patients with SSc hybrid cells coexpressing lymphatic vessel endothelial hyaluronan receptor-1 LYVE-1, a specific marker of lymphatic endothelial cells, and α-SMA was found exclusively in the fibrotic skin of SSc patients. The culturing of HdLy-MVECs with SSc serum or profibrotic TGFβ1 led to the acquisition of a myofibroblast-like morphofunctional phenotype, as well as the downregulation of lymphatic endothelial cell-specific markers and the parallel upregulation of myofibroblast markers [242]. The distribution of OSM receptor β (OSMRβ) was significantly increased in dermal EC and in fibroblasts of SSc patients compared to controls. In vitro treatment with OSM performed in human dermal endothelial cells, inducing cell migration and cell proliferation, proinflammatory cytokines, such as IL-6 and IL-33. The effects of OSM were mediated by OSMRβ and STAT3 SNAIL1, TGFβ3, ET-1, VE-cadherin and CD31, which also increased the levels of αSMA and TGFβ1, -2, -3 at (Table 4) [244]. 

Several studies underlined the role of promising molecules in EMT-dependent fibrosis and their potential interest as new therapeutic targets. Brachyury, a TF recently identified as an a potent inducer of EMT in human carcinoma cell lines [245], was overexpressed in SSc dermal fibroblasts both in vivo and in vitro compared to healthy controls. The silencing of brachyury reduced the expression of type I collagen in normal and SSc dermal fibroblasts, but did not decrease the levels of major disease-related cytokines [246]. Similarly, plasma levels of SPRT4-IT1, a lncRNA reported to be involved in EMT process in various cancers, was higher in diffuse than limited SSc and were positively correlated with modified Rodnan skin score (mRSS) [247]. In addition, E3 ubiquitin ligase was linked to EMT through the TGF-β SMAD-dependent signaling pathways [248]. Recently, LG283, a curcumin derivatives drug, exhibited a promising antagonistic activity on fibrosis and vascular injury in cultured human dermal fibroblasts and in the bleomycin-induced skin fibrosis mouse model through the inhibition of TGF-β/Smad/Snai1 pathway. Treatment with LG283 was able to downregulate the expression of Col1A2, α-SMA, FN, and phosphorylated Smad3 together with core EMT-TFs Snai1 and Snai2 [249]. Furthermore, several drugs also revealed encouraging results in that they attenuated fibrosis by hindering endoMT. Linagliptin, a specific dipeptidyl peptidase 4 (DPP4) inhibitor, alleviated pulmonary fibrosis in a bleomycin-induced mouse model of SSc. Treatment with linagliptin could mitigate the migration capacity of EC, decrease inflammatory cytokines, such as TNF-α and IL-6, and was a potent inhibitor of the Akt/mTOR pathway, resulting in higher levels of VE-cadherin and decreased levels of α-SMA, collagen I and Snai1 and Twist [250]. Macitentan, the antagonist of the endothelin-1 (ET-1) receptor, prevented endoMT and fibroblast accumulation in EC from SSc patients cultivated in vitro [241].

**Table 4 ijms-24-14481-t004:** Recapitulation table of findings regarding EMT/endoMT occurrence in SSc.

Cellular Type	Model	Signaling Pathway	Cytokine	TFs	Phenotypical Feature	Ref.
NHEK	Human/in vitro		TGF-β, TNF-α		spindle-shaped cells, ⇧ vimentin, ⇧ FN, ⇧ MMPs, ⇩ E-cadherin, ⇩ ZO-1	[231]
Skin biopsy (keratinocytes)	Human morphea		TGF-β1	SNAI1	⇧ α-SMA, ⇧ FN, ⇩ E-cadherin	[232]
Skin biopsy (keratinocytes)	Human dcSSc			SNAI1, TWIST1		[233]
Skin biopsy (keratinocytes)	Human dcSSc	TGF-β/SMAD		SNAI1	⇧ phosphorylated SMAD2/3	[230]
alveolar epithelial cells type II	Human/in vitro	TGF-β WNT/β-catenin			⇧ collagen type I, ⇧ vimentin	[234,237]
Keratinocytes	Human in vitro	WNT/β-catenin				[235]
Skin biopsy (keratinocytes)	Human	NOTCH				[236]
Skin biopsy (EC)	Human/in vitro	HIF-1α		HIF-1α	⇧ α-SMA, ⇧ VEGF-α	[240]
Skin biopsy (lymphatic EC)	Human/in vitro		TGFβ1		⇧ α-SMA, ⇩ LYVE-1	[242]
Skin biopsy (EC)	Human/in vitro	OSM/ OSMRβ/SNAI1	TGFβ3	SNAI1	⇧ ET-1, ⇧ α-SMA, ⇩ VE-cadherin	[244]
Dermal fibroblasts	Human/in vitro			Brachyury	⇧ collagen type I	[246]
Plasma	Human lcSSc			SPRT4-IT1	⇧ mRSS	[247]
Serum	Human dcSSc/ccSSc				⇩ MiR-138	[239]
Dermal fibroblasts	Huma in vitro + bleomycine mice	TGF-β/SMAD/SNAI1		SNAI1, SNAI2	⇧ phosphorylated SMAD3, ⇧ col1A2, ⇧ α-SMA, ⇧ FN	[249]
EC	bleomycine mice	Akt/mTOR pathway/Snai1	TNF-α, IL-6	Snai1, Twist	⇩VE-cadherin and ⇧ α-SMA, ⇧ collagen	[250]
EC	Human dcSSc	ET-1	TGF-β1		⇧ α-SMA, ⇧ col1A1	[241]

Abbreviations: α-SMA: α-smooth muscle actin; dcSSc: diffuse cutaneous SSc; EC: endothelial cells; EMT: epithelial-mesenchymal transition; endoMT: endothelial-mesenchymal transition; FN: fibronectin; lcSSc: Limited cutaneous SSc; LYVE-1: lymphatic vessel endothelial hyaluronan receptor-1 LYVE-1; MiR-138: Micro-RNA 138; MMPs: matrix metalloproteases; mRSS: modified Rodnan skin score; NHEK: normal human epidermal keratinocytes; SMAD: homolog of the Drosophila protein, mothers against decapentaplegic (Mad) and the Caenorhabditis elegans protein Sma; SSc: systemic sclerosis; TFs: transcription factors; TGF-β: transforming growth factor-β; TNF: tumor necrosis factor; TWIST1: Twist Family BHLH Transcription Factor 1; VE: vascular-endothelial; ZO-1: zona occludens 1; ⇧: upregulated; ⇩: downregulated.

## 4. Conclusions

In this review, the current knowledge about the entanglement of EMT in the pathogenesis of several autoimmune rheumatic disease was portrayed. Even though several lines of evidence seem to confirm the involvement of EMT in rheumatic diseases, its precise contribution to their pathophysiology remains unclear, and there are, unfortunately, lots of shortcomings and scant data in the literature. However, it seems that EMT contributes to some extent to the fibrogenic process stemming from rheumatic diseases. In addition, several proinflammatory cytokines have been linked to EMT detected in rheumatic diseases, suggesting that inflammation is an important prerequisite for triggering EMT. However, whether EMT can further contribute to disease progression or actually contribute to the pathophysiology of autoimmune diseases by interfering with the immune system still remains a big missing piece of the jigsaw. Fundamentally, further research is required to better comprehend its exact contribution to the physiopathology of rheumatic diseases and could pave the way for exciting new therapies.

## Figures and Tables

**Figure 1 ijms-24-14481-f001:**
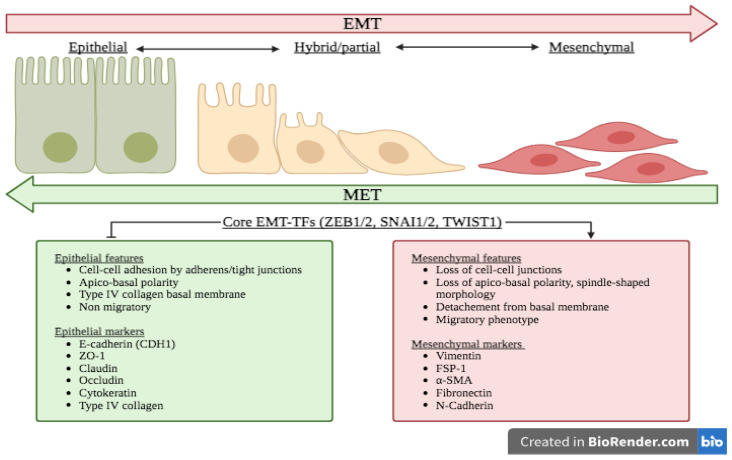
Morphological modifications during EMT. Epithelial cells are closely held together by different types of cell–cell junctions (i.e., adherens junctions and tight junctions) and form polarized sheets harboring an apico-basal polarity. The loss of E-cadherin (a key component of adherens junctions) is often considered the first event initiating EMT and is rapidly followed by the repression of other cell junction proteins such as zona occludens 1 (ZO-1), occludin, and claudin. Consequently, the dissolution of epithelial cell junctions elicits a loss of apico-basal polarity and triggers cytoskeletal modifications leading to the acquisition of a spindle-shaped morphology. The cells are pushed towards the EMT spectrum in a progressive and reversible manner, from a complete epithelial to complete mesenchymal state. Abbreviations: α-SMA: α-smooth muscle actin; CDH1: E-cadherin; EMT: epithelial–mesenchymal transition; FSP-1: fibroblast specific protein 1; MET: mesenchymal–epithelial transition; SNAI1: Snail family transcriptional repressor 1; TFs: transcription factors; TWIST1: twist family BHLH transcription factor 1; ZEB1: zinc finger E-box binding homeobox 1; ZO-1: zona occludens 1.

**Figure 2 ijms-24-14481-f002:**
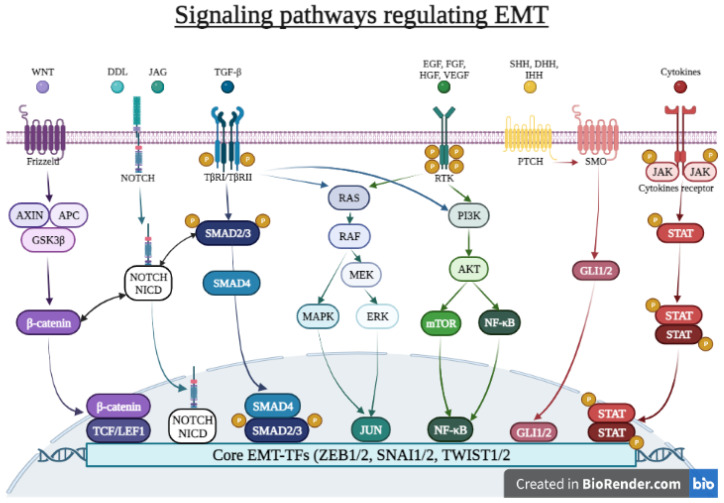
Signaling pathways regulating EMT. The encounter between specific ligands and epithelial cells results in the activation of several intracellular signaling pathways that ultimately lead to the expression of core EMT-TFs such as ZEB1/2, SNAI1/2, and TWIST1 that act pleiotropically to induce EMT. These pathways are not well compartmentalized, and some overlap exists to a certain extent between them. The canonical WNT pathway is activated upon the binding of WNT ligands to the Frizzled family of membrane receptors, ensuing the release of β-catenin from the GSK3β–AXIN-APC complex. This enables its translocation to the nucleus, thereby activating TCF and LEF-1 TFs, which promotes the expression of EMT-associated genes. The NOTCH pathway is activated upon binding of DDL or JAG ligands to the NOTCH receptor, leading to the release of the active NOTCH-ICD, which translocates to the nucleus to function as a transcriptional co-activator. The Hedgehog signaling pathway is activated by members of the Hedgehog (HH) ligand family that bind to PTCH receptor and activate SMO, which recruits GLI TFs, further entailing the transcription of EMT target genes. TGFβ activates the TGFβ family of receptors that trigger the phosphorylation and activation of cytoplasmic SMAD2 and SMAD3, which further assemble with SMAD4 to migrate to the nucleus, where they activate the transcription EMT-TFs. SMAD also interacts with β-catenin and NOTCH-ICD, enabling cross-talk between the TGF-β, WNT, and NOTCH pathways. The TGFβ pathway also collaborates with the PI3K–AKT pathway, which in turn triggers the activation of the mTOR and NF-κB and the RAS–RAF–MEK–ERK signaling axis. These pathways are also triggered by the binding of several growth factors to their cognate receptors. The binding of several cytokines to their receptors triggers the phosphorylation and activation of JAKs and STATs. STAT dimers activate the transcription of genes encoding core EMT-TFs. Abbreviations: AKT: protein kinase B; AXIN: axis inhibition protein; DHH: desert hh; DLL: Delta-like; EGF: epidermal growth factor; EMT: epithelial–mesenchymal transition; ERK: extracellular signal-regulated kinase; FGF: fibroblast growth factor; GSK-3β: glycogen synthase kinase-3β; HGF: hepatocyte growth factor; IHH: Indian Hedgehog; JAG: jagged; JAK: Janus kinase; LEF-1: lymphoid enhancer binding factor 1; MAPK: mitogen-activated protein kinase; MEK: MAP kinase; mTOR: mammalian target of rapamycin; NF-κB: nuclear factor-κB; NOTCH-ICD: NOTCH receptor intracellular domain; PI3K: phosphoinositide 3-kinase; PTCH: patched; RTKs: receptor tyrosine kinases; SHH: sonic hh; SMAD: homolog of the Drosophila protein, mothers against decapentaplegic (Mad) and the Caenorhabditis elegans protein Sma; SMO: smoothened; SNAI1: snail family transcriptional repressor 1; STAT: signal transducers and activators of transcription; TβRI: TGFβ receptor type I; TβRII: TGFβ receptor type II; TCF: T cell factor; TFs: transcription factors; TGF-β: transforming growth factor-β; TWIST1: twist family BHLH transcription factor 1; VEGF: vascular endothelial growth factor; ZEB1: zinc finger E-box binding homeobox 1.Created with Biorender.com (accessed on 20 August 2023).

**Figure 3 ijms-24-14481-f003:**
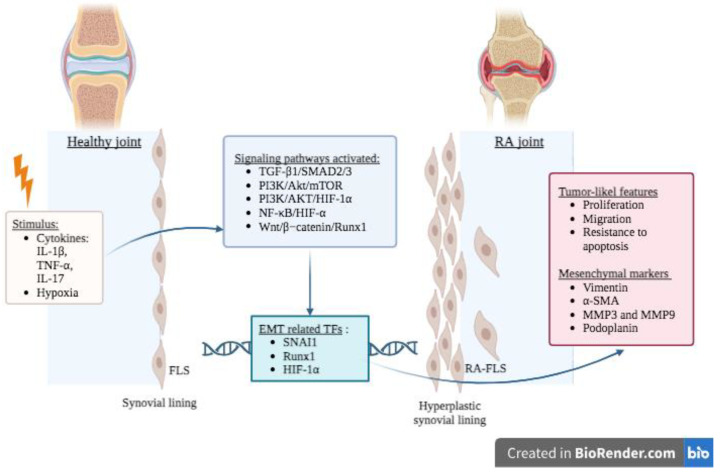
Hypothetical contribution of EMT-like process to the pathophysiology of RA. Upon stimuli such as inflammation (IL-1β, TNF-α and IL-17) or hypoxia, several signaling pathways become activated in normal FLS present in the synovial lining. These EMT signaling pathways culminate with the production of several EMT-related TFs such as SNAI1, HIF-1α and Runx1 leading to the acquisition of more mesenchymal features. These activated FLS called RA-FLS gain migratory and invasive properties together with an increased resistance to apoptosis resulting in the generation of hyperplastic synovial lining. Abbreviations: AKT: protein kinase B; α-SMA: α-smooth muscle actin; CIA: collagen-induced arthritis; EMT: epithelial-mesenchymal transition; HIF1α: hypoxia-inducible factor 1α; IL: interleukin; MMPs: matrix metalloproteases; mTOR: mammalian target of rapamycin; NF-κB: nuclear factor-κB; PI3K: phosphoinositide 3-kinase; RA: rheumatoid arthritis; RA-FLSs: RA fibroblast-like synoviocytes; RUNX1: runt-related transcription factor 1; SMAD: homolog of the Drosophila protein, mothers against decapentaplegic (Mad) and the Caenorhabditis elegans protein Sma; SNAI1: Snail family transcriptional repressor 1; TFs: transcription factors; TGF-β: transforming growth factor-β; TNF: tumor necrosis factor. Created with Biorender.com (accessed on 20 August 2023).

**Figure 4 ijms-24-14481-f004:**
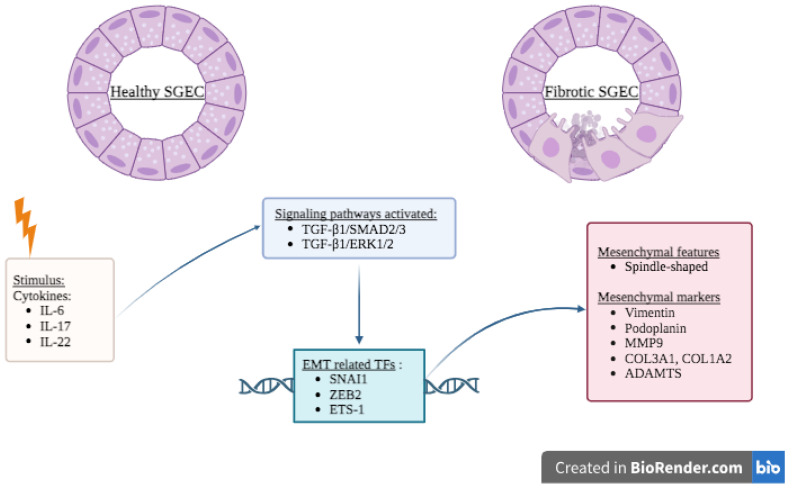
Hypothetical contribution of EMT to the pathophysiology of pSS. Upon stimuli such as inflammation (IL-6, IL-17 and IL-22), several signaling pathways become activated in normal SGEC present in salivary glands. These EMT signaling pathways culminate with the production of several EMT-related TFs such as SNAI1, ZEB1 and ETS1 leading to the acquisition of mesenchymal features. This results in salivary gland fibrosis with the appearance of spindle-shaped morphology of SGEC together with the expression of mesenchymal markers such as vimentin, podoplanin, MMP9 and several collagen types. Abbreviations: ADAMTS: A Disintegrin And Metalloproteinase with Thrombospondin Motifs; COL3A1: collagen type III Alpha 1 chain; EMT: epithelial-mesenchymal transition; ERK: extracellular signal–regulated kinase; IL: interleukin; iSGEC: immortalized SGEC; MMPs: matrix metalloproteases; SGEC: salivary gland epithelial cells; SMAD: homolog of the Drosophila protein, mothers against decapentaplegic (Mad) and the Caenorhabditis elegans protein Sma; SNAI1: Snail family transcriptional repressor; SS: Sjögren’s syndrome; TFs: transcription factors; TGF-β: transforming growth factor-β; ZEB2: zinc finger e-box binding homeobox 2. Created with Biorender.com (accessed on 20 August 2023).

**Table 1 ijms-24-14481-t001:** Recapitulation table of findings regarding EMT-like occurrence in RA.

Cellular Type	Model	Signaling Pathway	Cytokine	TFs	Phenotypical Feature	Ref.
RA-FLS	human/in vitro				tumor-like	[165,166,167,168,169]
RA-FLS	human/in vitro				⇧ α-SMA	[31,171,172]
RA-FLS	human/in vitro	TGF-β1	IL-1β, TNF-α		⇧ podoplanin	[172]
RA-FLS	human/in vitro	TGF-β1/SMAD2/3		SNAI1	invasion	[31]
RA-FLS	human/in vitro	PI3K/AKT/HIF-1α		HIF-1α	invasion	[165]
RA-FLS	human/in vitro	NF-κB/HIF-α	IL-17		⇧ MMP2, ⇧ MMP9, invasion	[44,175]
RA-FLS	in vitro SW982 human cell line	PI3K/Akt/mTOR		SNAI1	⇧ vimentin, ⇧ MMP2, ⇧ MMP9, ⇩ E-cadherin	[177]
RA-FLS	human/in vitro + CIA rats	NOTCH2/3		HIF-α	invasion, resistance to apoptosis	[176]
RA-FLS	CIA rats	Pi3k-/Akt		Snai1	invasion	[178]
RA-FLS	CIA rats	Wnt/β−catenin		Runx1	invasion	[170]

Abbreviations: AKT: protein kinase B; α-SMA: α-smooth muscle actin; CIA: collagen-induced arthritis; EMT: epithelial-mesenchymal transition; HIF1α: hypoxia-inducible factor 1α; IL: interleukin; MMPs: matrix metalloproteases; mTOR: mammalian target of rapamycin; NF-κB: nuclear factor-κB; PI3K: phosphoinositide 3-kinase; RA: rheumatoid arthritis; RA-FLSs: RA fibroblast-like synoviocytes; Ref.: references; RUNX1: runt-related transcription factor 1; SMAD: homolog of the Drosophila protein, mothers against decapentaplegic (Mad) and the Caenorhabditis elegans protein Sma; SNAI1: Snail Family Transcriptional Repressor 1; TFs: transcription factors; TGF-β: transforming growth factor-β; TNF: tumor necrosis factor; ⇧: upregulated; ⇩: downregulated.

**Table 2 ijms-24-14481-t002:** Recapitulation table of findings regarding EMT occurrence in pSS.

Cellular Type	Model	Signaling Pathway	Cytokine	TFs	Phenotypical Feature	Ref.
SGEC	human/in vitro	TGF-β1/SMAD		SNAI1	spindle-shaped, ⇧ vimentin, ⇧ collagen type I, ⇩ E-cadherin	[33]
SGEC	human/in vitro	TGF-β1	IL-6		spindle-shaped, ⇧ vimentin, ⇧ collagen type I, ⇩ E-cadherin	[181]
SGEC	human/in vitro		IL-17, IL-22		spindle-shaped, ⇧ vimentin, ⇧ collagen type I, ⇩ E-cadherin	[182]
SGEC	human/in vitro				podoplanin	[193]
SGEC	human/in vitro	TGF-β1/SMAD2/3 TGF-β1/ERK1/2	IL-17		spindle-shaped, ⇧ vimentin, ⇧ collagen type I, ⇩ E-cadherin	[184]
SGEC/iSGEC	human/in vitro			ETS1	⇧ MMP9	[195]
SGEC	human/in vitro			ZEB2	⇧ COL3A1, COL1A2 and ADAMTS	[196]

Abbreviations: ADAMTS: A Disintegrin And Metalloproteinase with Thrombospondin Motifs; COL3A1: collagen type III Alpha 1 chain; EMT: epithelial-mesenchymal transition; ERK: extracellular signal–regulated kinase; IL: interleukin; iSGEC: immortalized SGEC; MMPs: matrix metalloproteases; Ref.: references; SGEC: salivary gland epithelial cells; SMAD: homolog of the Drosophila protein, mothers against decapentaplegic (Mad) and the Caenorhabditis elegans protein Sma; SNAI1: Snail Family Transcriptional Repressor 1; SS: Sjögren’s syndrome; TFs: transcription factors; TGF-β: transforming growth factor-β; ZEB2: Zinc Finger E-Box Binding Homeobox 2; ⇧: upregulated; ⇩: downregulated.

**Table 3 ijms-24-14481-t003:** Recapitulation table of findings regarding EMT or endoMT occurrence in SLE.

Cellular Type	Model	Signaling Pathway	Cytokine	TFs	Phenotypical Feature	Ref.
glomeruli/TEC	human LN biopsy		TGF-β1/β3			[207,208]
mesangial cells	human LN biopsy		TGF-β1		⇧ FN	[197]
TEC	human/in vitro		Th17/IL-17A		⇧ FN, ⇩ E-cadherin	[210,211]
TEC	human/in vitro		IL-17A, TGF-β1		⇧ collagen I and III, ⇧ α-SMA ⇩ E-cadherin	[210]
serum	human LN		TGF-β1		⇧ miR-130b-3p, ⇧ α-SMA, ⇩ E-cadherin	[212]
lung	Human/lung biopsy SLE	NETs		Twist1, Snai1	⇧ α-SMA, ⇩ E-cadherin	[213]
Glomeruli/EC	human LN biopsy/in vitro EC/ MRL/Lpr mice	NETs/β-catenin/Snai1		Snai1	⇧ α-SMA, ⇧ MMP9, ⇩ VE-cadherin	[214]
TEC	MRL/Lpr mice	Jak/Stat/OSM	OSM	Stat1/Stat3	⇧ α-SMA, ⇧ FN, ⇩ E-cadherin	[215]
TEC	MRL/lpr mice	Wnt/β-catenin	FKN	β-catenin	⇧ α-SMA, ⇧ vimentin, ⇧ Wnt-4	[216]
TEC	MRL/lpr mice	Stim1			⇧ FN, ⇧ urine protein, ⇧ serum creatinine	[217]
TEC	MRL/lpr mice	Yap1			⇧ FN, ⇧ renal injury	[218]
TEC	MRL/lpr mice/in vitro		TGF-β1		⇧ FN, ⇩ E-cadherin	[219]
TEC	MRL/lpr mice	TGF-*β*1/Smad2/3			⇧ α-SMA, ⇧ phosphorylated SMAD2/3, ⇩ E-cadherin	[220]

Abbreviations: α-SMA: α-smooth muscle actin; EC: endothelial cells; EMT: epithelial–mesenchymal transition; endoMT: endothelial-mesenchymal transition; FN: fibronectin; IL: interleukin; JAK: janus kinase; LN: lupus nephritis; NET: neutrophil extracellular traps; OSM: oncostatin M; Ref.: references; SLE: systemic lupus erythematosus; SMAD: mothers against decapentaplegic homolog; SNAI1: Snail family transcriptional repressor 1; STAT: signal transducers and activators of transcription; TEC: tubular–epithelial cell; TFs: transcription factors; TGF-β: transforming growth factor-β; TWIST1: twist family BHLH transcription factor 1.; VE: vascular-endothelial; ⇧: upregulated; ⇩: downregulated.

## Data Availability

Not applicable.

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
