# Peer review of "Involvement of Epithelial-Mesenchymal Transition (EMT) in Autoimmune Diseases"

_ijms, 2023, doi:10.3390/ijms241914481_

Round 1

Reviewer 1 Report

In the review entitled “Involvement of epithelial-mesenchymal transition (EMT) in autoimmune diseases” the authors report that it is well accepted that in fibrosis, characterized by an excessive amount of myofibroblasts and an abundant production of extracellular matrix (ECM) proteins, a significant part of myofibroblasts results from the conversion of epithelial cells through EMT. At the same, many chronic autoimmune and inflammatory diseases, such as systemic sclerosis (SSc), rheumatoid arthritis (RA), systemic lupus erythematosus (SLE), Sjögren’s syndrome (SS) and inflammatory bowel diseases (IBD) are related to EMT. The authors also wrote that chronic inflammatory state observed in most autoimmune and inflammatory diseases can act as a potent trigger of EMT, leading to the development of a pathological fibrotic state. However, this intriguing  hypotheis is not well confirmed in literature. In fact, as the authors wrote in the Introduction section, theEpithelial-to-mesenchymal transition (EMT) is a complex reversible biological process that takes place in epithelial cells and is characterized by the loss of epithelial and the acquisition of mesenchymal features.” (see Figure 1), and therefore this process should be applied only to the epithelial cells and not the mesenchymal ones, like the authors often reported in some crhonic diseaeses. Some examples:

In 3.1. Rheumatoid arthritis:  the authors reported the sentence; “Over the past 15 years, several lines of evidence seem to indicate that RA-FLS may, at least in part, undergo EMT (159, 160).” However the paper 160 did not cite any EMT of synoviocytes. And the authors of the paper 159 who found collagen IV only suggested that the synovicytes display an epithelial-like phenotype. There are not epithelial cells in sinovium that includes stromal cells. How these cells can detach from the basement membrane (Figure 1). In fact in Smith, 2011 and other papers deny the presence of a basement membrane and clearly wrote that even though they detected normal component of a basement membrane, “there is no basement membrane beneath the intimal layer in normal synovium” (from Smith MD. The normal synovium. Open Rheumatol J. 2011;5:100-6. doi: 10.2174/1874312901105010100. Epub 2011 Dec 30. PMID: 22279508; PMCID: PMC3263506.)

In 3.3. Systemic lupus erythematosus the authors wrote: “Interestingly, several studies demonstrated that 5-36% of the interstitial a-SMA positive matrix-producing myofibroblasts observed in human renal biopsies originated from tubular epithelial cells (29, 191).” Only the paper 29 clearly reported that some fibroblasto derived from an EMT process causing fibrosis. From the cited paper 191, “This study confirms previous reports that EMT program does not lead to the generation of a significant number of interstitial myofibroblasts in fibrosis” and “only about 5% of myofibroblasts are truly derived via a fully executed EMT. “

Moreover,the MMPs are briefly cited, but these molecules play a fundamental role in developing the fibrosis process and also characterize the EMT process which allows epithelial-mesenchymal cell migration.

This subject is very interesting but the authors should improve the review.

Minor points:

 Page 2, line 10: The authors previously report the involment of EMT in injury and wound which may be considered pathological processes. Therefore the suggested addition would be usefull.Change “triggered upon pathological condition” in “triggered upon more dramatic pathological condition”.

Page 2, line 13: Change “intravasation to the blood.“  in "intravasation to the blood and lymphatic vessels”.

Page 2, last sentence of paragraph 2: Change “to the repression of epithelial genes and the activation of mesenchymal genes accounting for the loss of epithelial and the gain of mesenchymal traits observed during EMT” in “to the repression of epithelial genes and the activation of mesenchymal ones accounting for the loss of epithelial and the gain of mesenchymal traits.”

Page 3, line 7: “N-cadherin” is remote twice. Delete one.

Page 3, Fig.1: In Epithelial markers, write markers as singolar and not plural. In Mesenchymal features, correct “Detachement of basement membrane” in “Detachment from the basement membrane”.

Page 4, line 1: the sentence “Some of these were identified as master regulators or core EMT-TFs as they are conserved across various biological conditions and across species, and others may have more restricted functions depending on the tissue or the biological background (40, 41), suggesting that the transcriptional regulation of EMT involves multiple layers of control (20, 41).” Is too long. Put a full stop and trnasform it into two sentences.

Page 4, line 5 of Paragraph 2.2.1: “including genes encoding claudins and occludins,” Report them at singolar.

Page 4, line 8 of Paragraph 2.2.1: “to induce MMPs expression”. Please briefly expalin what they are and do.

Page 4, line 3 of Paragraph 2.2.3: “a mesenchymal gene   (57)”. Reduce the free space.

Page 4, line 2 of Paragraph 2.2.4: Reduce the free space.

Page 5, at the end of Parapraph 2.3.1: Change “the activation of mesenchymal genes and the repression of epithelial genes.” In “the activation of mesenchymal genes and the repression of epithelial ones.”

Page 5, line 1 of Paragraph 2.3.3: Change “pathways”  in “pathway”.

Page 8:, title: Change “3. EMT in various rheumatic conditions.” in “3. EMT in various rheumatic diseases”.

Page 8, at the end of paragraph 2.4.1: “IBD and IPF”. Please specify the diseases.

Page 8, line 2 of paragraph 2.4.3: Change “ECM influence EMT” in ECM influences EMT.

Page8, last sentence of paragraph2.4.3: “In addition, MMP-3 has also been shown to directly induce EMT through activation of Rac1 GTPase–reactive oxygen species (ROS) signaling, which promotes expression of SNAI1 (143).” MMP3 is not a component of ECM. Delete this sentence or modify to better explain.

Page 8, line 2 of paragraph 3,1: “joints  inflammation”. Delete the free space.

Reviewer 2 Report

Authors should revise the manuscript accordingly.

Minor corrections are needed.

Round 2

Reviewer 1 Report

The first half part of this paper is focused on the general definitions and characteristics of epithelial-to-mesenchymal-transition (EMT). Only the second half part is specifically concerning “The Involvement of epithelial-mesenchymal-transition (EMT) in autoimmune diseases” (indeed only three of these diseases might bereally involved: the Sjögren’s syndrome, Systemic lupus erythematosus, and Systemic sclerosis. I sincerely appreciate the efforts of the authors to improve their paper, but also reading other published studies I confirm that value of words is strongly related to their meaning. If we speak about EMT we must demonstrate a transition from an epithelial to a mesenchymal phenotype. In literature, when mesothelial cells, which even though act like epithelial cells are not epithelial, change their phenotype to become mesenchymal the other authors consider a mesothelial-mesenchymal-transition (MMT). I repeat that in rheumatoid arthritis the RA-FLS cells reported in all literature studies in paragraph 3.1 and Table 1 are fibroblast-like synoviocites and not epithelial cells. There is no transition in a mesenchymal phenotype which become “more” mesenchymal. At the same the mesangial cells, cited in paragraph 3.3 and Table 3, originate from mesenchymal or stromal cells and are not epithelial. I am sorry to repeat my personal considerations which justified my previous opinion. 

Round 3

Reviewer 1 Report

I read the efforts of the authors but some important points still remain and the paper should be rewritten:

1) As I wrote in the previous report, there is no proportion between the first part of 9 pages (concerning the only generic EMT parameters) and the second one of 10 pages with the included large Tables) concerning the real subject of the paper.

2) The epithelial cells originate from ectoderm while the synovial membrane is a connective tissue with synovial cells (Type A resembling macrophage, and type B resembling fibroblasts) deriving from mesoderm. One thing is the EMT process and something else the EMT markers which are the only one which increase in RA.

3) The increase of gene expression of EMT markers in RA and EMT process in the other autoimmune diseases is not a surprise because EMT is reported in development, wound, fibrosis and cancer: i.e. everywhere there is tissue remodeling. For this reason I suggested to introduce in the review the proteolytic enzimes (MMPs) which digest the extracellular matrix and are related to EMT. In view of this  concept, the EMT is not a particular characteristic of autoimmune diseases, but is related to tissue rebuilding. The authors should also widely and clearly explain this to avoid confusion in the readers.

4) The aim of a review should be to summarize researches concerning a common subject (in this case the autoimmune diseases), clarify, discuss and compare the common and contrasting results previously reported to clarify the content to the readers. The overgeneralization in this manuscript causes confusion in the readers.

Author Response

Dear Editor,

We thank the reviewer for their helpful comments with a view to improving the quality of the manuscript.

We stress on the fact that the manuscript is a review on the Emerging role of EMT in autoimmune diseases.

The first part of our manuscript describes the different major mechanisms underlying EMT and has been tailored in a way that the “ reader”  can get a full grasp of this yet unknown entity.

The second part of the manuscript describes the current literature on EMT and different inflammatory and auto-immune diseases. Even if there has been a quantum leap in research in deciphering the role of EMT in these inflammatory and auto immune processes , there is unfortunately, lots of shortcomings and scant data. 

Regarding the comments of the reviewer no 1 , we thank him a lot for his comments  but cannot agree dully what has been proposed.  The manuscript has been formatted in a way that every person who is not an expert of EMT can understand its function and downstream implications.

Regarding  query no 2:

This point has been addressed and changed accordingly throughout the text. In the current literature, from a strict point of view, there are several lines of evidence corroborating the fact that EMT is not restricted to epithelial cells per se , but has been extensively proven in other cell types such as non-epithelial cancer cells, endothelial cells or mesothelial cells of pulmonary fibrosis. From this stance, these authors concluded that as such , the definition of  EMT should be stretched to the involvement of other cell types. This stance point has been added in the text and helps in unshrouding any further shadows of doubt.

Regarding query no 3:

We cannot agree with the statement in the fact that it is purely based on the opinion of the reviewer and objectively and scientifically proven .

We have added the role of MMP in the remodeling of the extracellular matrix in the text .

I do hope that the reviewer takes our remark into account and understand our point of view.

Regarding query 4 :

It is related to the pure subjective opinion of the reviewer. We have strived to resume in the best way possible what has been reported in the literature. There is to our sense no overgeneralization but strictely facts and data that have been reported.

Kind Regards,

Prof Soyfoo and Dr Sarrand
